# Characterization of a Novel L-Asparaginase from *Mycobacterium gordonae* with Acrylamide Mitigation Potential

**DOI:** 10.3390/foods10112819

**Published:** 2021-11-16

**Authors:** Huibing Chi, Meirong Chen, Linshu Jiao, Zhaoxin Lu, Xiaomei Bie, Haizhen Zhao, Fengxia Lu

**Affiliations:** College of Food Science and Technology, Nanjing Agricultural University, Nanjing 210095, China; 2019208015@njau.edu.cn (H.C.); chenmr@njau.edu.cn (M.C.); 2016208023@njau.edu.cn (L.J.); fmb@njau.edu.cn (Z.L.); bxm43@njau.edu.cn (X.B.); zhaohz@njau.edu.cn (H.Z.)

**Keywords:** L-asparaginase, acrylamide, *Mycobacterium gordonae*, thermal stability

## Abstract

L-asparaginase (E.C.3.5.1.1) is a well-known agent that prevents the formation of acrylamide both in the food industry and against childhood acute lymphoblastic leukemia in clinical settings. The disadvantages of L-asparaginase, which restrict its industrial application, include its narrow range of pH stability and low thermostability. In this study, a novel L-asparaginase from *Mycobacterium gordonae* (GmASNase) was cloned and expressed in *Escherichia coli* BL21 (DE3). GmASNase was found to be a tetramer with a monomeric size of 32 kDa, sharing only 32% structural identity with *Helicobacter pylori* L-asparaginases in the Protein Data Bank database. The purified GmASNase had the highest specific activity of 486.65 IU mg^−1^ at pH 9.0 and 50 °C. In addition, GmASNase possessed superior properties in terms of stability at a wide pH range of 5.0–11.0 and activity at temperatures below 40 °C. Moreover, GmASNase displayed high substrate specificity towards L-asparagine with Km, kcat, and kcat/Km values of 6.025 mM, 11,864.71 min^−1^ and 1969.25 mM^−1^min^−1^, respectively. To evaluate its ability to mitigate acrylamide, GmASNase was used to treat potato chips prior to frying, where the acrylamide content decreased by 65.09% compared with the untreated control. These results suggest that GmASNase is a potential candidate for applications in the food industry.

## 1. Introduction

L-asparaginase is a hydrolase that can catalyze the conversion of L-asparagine to L-aspartic acid and ammonia [1]. This enzyme has attracted a great deal of attention in clinical settings as a valuable drug in the treatment of acute lymphoblastic leukemia. Additionally, it has a great potential as a promising agent for producing acrylamide-free food [2,3]. Acrylamide is a dangerous carcinogen that is produced in the processing of carbohydrate-based foods at temperatures above 120 °C and low moisture conditions [4]. By hydrolyzing L-asparagine (the crucial precursor substance of acrylamide), L-asparaginase could effectively mitigate the formation of acrylamide via Maillard reaction by 55–96% [5,6,7,8,9,10,11,12]. Hence, the production of L-asparaginase with desired properties has attracted a great deal of interest in terms of medical therapy and food safety.

L-asparaginase has been found in animals, plants and microorganisms. However, only microbial L-asparaginases are of commercial significance due to their being highly efficient and inexpensive, properties that have facilitated industry-scale production [13]. To date, the properties of L-asparaginases from a variety of microorganisms have been characterized [13,14], especially the L-asparaginases from *Erwinia, Pseudomonas sp, Bacillus, Aspergillus*, and *Escherichia coli* [15,16]. Unfortunately, commercially approved L-asparaginases that are used in the food industry for acrylamide mitigation are only from *Aspergillus niger* and *Aspergillus oryzae* [16]. The species is rare and the price is expensive. Moreover, *A. oryzae* and *A. niger* L-asparaginases were only stable in the pH ranges of pH 6.0–9.0 and pH 5.0–9.0, respectively [6,17,18], which could not meet low or high pH demand in food processing [14]. In addition, the optimum temperature of *A. niger* L-asparaginase was only 30 ℃, which would limit its application in thermal food processing. In short, its narrow pH range stabilization and low thermal stabilization have a restrictive effect on its industrial application. Considering the demand for a high specificity, high thermostability, and wide pH range stability of L-asparaginase in industrial applications, one alternative could be to use other microbial sources of L-asparaginase in order to alleviate these problems [19,20].

Actinomycetes that are closely related to human beings, which can produce abundant bioactive molecules and a variety of enzymes, have the ability to provide L-asparaginase with better properties compared to bacterial and fungal species [15,21]. However, L-asparaginases derived from actinomycetes are rarely reported [22]. To date, L-asparaginases from actinomycetes have only been focused on *Streptomyces* such as *S. ginsengisoli* [23], *S. phaeochromogenes* [24], *S. fradiae* [25], and *S. gulbargensis* [26], and their potential for acrylamide mitigation in the thermal processing of food has not been investigated.

In this study, we successfully achieved the heterologous expression of a novel L-asparaginase from actinomycete *Mycobacterium gordonae* in *E. coli* BL21 (DE3). Additionally, the properties of the recombinant enzyme were characterized. Furthermore, we also evaluated the reduction in acrylamide in fried potato chips treated with the recombinant L-asparaginase.

## 2. Materials and Methods

### 2.1. Strains and Reagents

The L-asparaginase coding gene (accession number WP_069434439) was synthesized and cloned to the plasmid pET-30a (+)-gm by Genewiz. *E.coli* BL21 (DE3) was preserved in our laboratory. L-asparagine, D-asparagine, L-glutamine, L-aspartic acid, D-aspartic acid, L-glutamate, mercuric potassium iodide, and trichloroacetic acid were purchased from Aladdin (Shanghai, China). Kanamycin and isopropyl β-D-thiogalactopyranoside (IPTG) were obtained from Solarbio (Beijing, China). All other chemicals were of analytical grade.

### 2.2. In Silico Sequence and Phylogenetic Analysis

The nucleotide and amino acid sequences obtained were compared against database sequences using NCBI BLAST, and MEGA 7.0 was used to calculate the identities of the nucleotide and amino acid sequences. The molecular masses and theoretical pI values of the gm were predicted using the ProtParam tool of Expasy.

### 2.3. Structural Modeling of Mycobacterium Gordonae L-Asparaginase and Structure Analysis

Secondary structure prediction was performed using the PSIPRED program. The three-dimensional (3D) modelling for the deduced amino acid sequences of L-asparaginase was performed using the Swiss-Model server. The 3D structure of *Mycobacterium gordonae* L-asparaginase was analyzed with PyMOL [27].

### 2.4. Expression and Purification of Mycobacterium Gordonae L-Asparaginase in Escherichia coli

*E. coli* BL21 (DE3) harboring plasmid containing the GmASNase gene was inoculated at 37 °C and an agitation rate of 180 rpm until an OD_600_ of 0.6–0.8 was reached. Protein expression was induced by adding IPTG at a final concentration of 100 mg mL^−1^, followed by incubation at 16 °C for 20 h [10,28]. The cells were collected by centrifugation at 8000× *g* for 5 min, before being suspended with Tris-HCl buffer (20 mM Tris-HCL, 300 mM NaCl, pH 8.0). Then, the suspension was sonicated at 200 W for 10 min and cell debris was removed by centrifugation (10,000× *g*, 30 min) at 4 °C. The supernatant was collected and loaded onto Ni-NTA column for purification. The purity and integrity of recombinant GmASNase was checked by SDS-PAGE (5% (*w/v*) polyacrylamide stacking gel and 12% (*w/v*) separation gel). The recombinant GmASNase (3μg/μL) was added into 4× protein loading buffer solution, and was then placed in a boiling water bath for 3 –5 min. The bands were stained with Coomassie Brilliant Blue R250 and decolorized with a mixed aqueous solution of 10% ethanol and 10% acetic acid for easy observation [10,28].

### 2.5. Protein Determination and Asparaginase Activity Assay

The protein concentration was determined by the Bradford method using bovine serum albumin as the standard [29]. The L-asparaginase activity was detected using the Nessler measurement with Nessler’s reagent according to Kumar et al. [30]. There were two steps: firstly, 700 μL of PBS buffer (50 mM, pH 8.0), 100 μL of 200 mM L-asparagine, and 100 μL of suitably diluted enzyme were mixed and incubated at 37 ℃ for 10 min and then terminated by adding 100 μL of 25% trichloroacetic acid solution. Secondly, 40 μL of the above reaction supernatant, coupled with 100 μL of Nessler’s reagent, was mixed with 860 μL of deionized water as the samples to be detected. After 5 min, the amount of ammonia released was determined by OD_436_ at room temperature.

### 2.6. Effects of Temperatures and pHs on GmASNase Activity and Stability

The optimum temperature of GmASNase was measured by assessing L-asparaginase activity at various temperatures (25–80 °C). The maximum activity of L-asparaginase within the aforementioned temperature range was designated as 100% in order to calculate the relative activity at each temperature point. Thermostability was measured by determining the residual enzyme activity of the purified enzyme after incubation at different temperatures (4–50 °C) for 50 min before a 30 min ice bath process. Enzyme activity at time = 0 was defined as a 100% activity for comparisons.

The catalytic activity of GmASNase was detected from pH 4.0 to 12.0 in order to determine its optimum reaction pH, the involved buffers were listed as follows [5,31,32]: pH 4.0–6.0, citrate acid-sodium citrate buffer; pH 6.0–9.0, phosphate-buffer solution; pH 9.0–10.0, glycine-sodium hydroxide buffer; pH 10.0–11.0, sodium carbonate-sodium hydroxide buffer; pH 11.0–12.0, disodium bicarbonate-sodium hydroxide buffer. We defined the highest activity of L-asparaginase as 100% in order to calculate the relative activity at each pH value. Furthermore, for the pH stability, GmASNase was incubated in buffers with different pH values at 4 °C for 24 h and residual activity was determined under assay conditions (pH 9.0) [33,34,35]. 

### 2.7. Effect of Metal Ions and Inhibitors on GmASNase Activity

The effects of metal ions and inhibitors on GmASNase activity were determined by the addition of 1 mM of various metal ions (KCl, NaCl, CaCl_2_, MgCl_2_, ZnCl_2_, CuSO_4_, FeSO_4_, MnCl_2_, FeCl_3_, AlCl_3_, and NiSO_4_) and inhibitors (sodium dodecyl sulfate (SDS) and ethylene diamine tetraacetic acid (EDTA)). The activity of GmASNase without adding any metal ions or inhibitors was set as 100%.

### 2.8. Substrate Specificity Assay and Determination of Kinetic Parameters of GmASNase

Substrate specificity was determined using different substrates and products at a final concentration of 20 mM such as L-asparagine (L-ASN), D-asparagine (D-ASN), L-glutamine (L-GLN), L-aspartic acid (L-ASP), D-aspartic acid (D-ASP), and L-glutamic acid (L-GLU). The activity was determined by the Nessler assay, as described above. The enzyme activity toward L-asparagine was set as 100%. Kinetic parameters were determined by incubating the recombinant enzyme with a substrate concentration in the range of a final concentration of 1–20 mM (50 mM PBS buffer pH 8.0) at 37 °C for 2 min. The Km and Kcat values were calculated using the Michaelis–Menten equation from the Lineweaver–Burk plots. 

### 2.9. Application of GmASNase in Food

#### 2.9.1. Preparation of Potato Chips 

Potato chips were prepared according to the method of Lu [28], with some modifications. The potatoes were washed, peeled, and cut into 2 mm pieces. Additionally, the starch particles adsorbed on the surface of the potatoes were then rinsed with ultra-pure water. The potato chips were immersed in 40 IU mL^−1^ [6,8,11] crude enzyme solution at 37 °C for 30 min. The potato chips were dried at 60 °C for 20 min, fried at 180 °C for 5 min, and then cooled to room temperature. 

#### 2.9.2. Separation and Purification of Acrylamide in the Sample

The extraction of acrylamide from potato chips was implemented according to the method of Jiao [10], with some modifications. The fried potato chips were mashed, homogenized, and weighed (10 g) in a centrifuge tube, and degreased with 50 mL of n-hexane three times. Additionally, 25 mL of acetonitrile, 25 mL of ultra-pure water, 5 g of NaCl, and 20 g of magnesium sulfate were then added to the potato chip samples, and the mixtures were subjected to ultrasound for 30 min and centrifuged at 10,000× *g* for 10 min. The acetonitrile layer was collected and rotated for evaporation, before being redissolved with 1 mL of ultrapure water. Finally, resuspended liquids were filtered with a 0.22 mm microporous membrane for further analysis. 

#### 2.9.3. Detection of Acrylamide

The acrylamide quantitative analysis was determined by the liquid chromatography-tandem mass spectrometry method (LC-MS/MS) using a Kinetex ^®^ F5 100 A (2.1 × 100 mm, 2.6 μm) chromatographic column. The samples were eluted at a 0.25 mL min^−1^ flow rate with a mixture of 90% water and 10% methanol (*v/v*, isometric elution). Acrylamide was detected at ions m/z 72.1 and m/z 54.9 in electrospray ionization source positive ion mode.

### 2.10. Statistical Analysis

The experimental results were expressed as the means and standard deviations (SDs) of the replicate determinations. The experimental results all came from the same day, and the same analyst. All statistical analyses were carried out using Origin 2021.

## 3. Results

### 3.1. In Silico Sequence and Phylogenetic Analysis

The open reading frame (ORF) of GmASNase gene encoded for 312 amino acids with the calculated molecular mass of 31.308 kDa and the theoretical pI of 5.27. The nucleotide and amino acid sequences of GmASNase were analyzed with sequences of *Erwinia chrysanthemi* L-asparaginase (GenBank accession number CAA32884), *Streptomyces radiopugnans* MS1 L-asparaginase (JF799106), *Nocardiopsis alba* NIOT-VKKMA08 L-asparaginase (KF724082), *Mycobacterium gordonae* L-asparaginase (WP_205877843), *Nocardiopsis dassonvillei* subsp. L-asparaginase (CP002040), and *Mycobacterium vicinigordonae* QLL09639.1 L-asparaginase (CP059165). Analysis of amino acid sequences in Figure 1B revealed that GmASNase belonged to L-asparaginase and the highest percentage of identity was observed with WP_205877843. However, its nucleotide sequences were much different from other species (Figure 1A). Bacterial species switched to different clusters for L-asparaginase genes at both the nucleotide and amino acid levels, indicating divergence among the organisms. 

### 3.2. Structural Modeling of Mycobacterium Gordonae L-Asparaginase and Structure Analysis

Using the PSIPRED program, the secondary structure of GmASNase was predicted to have a maximum of eleven α-helix and thirteen β-strands with maximum hydrophilic molecules (Figure 2A). Moreover, a large number of regions were also predicted to have favorable formations of coils in the deduced amino acid sequences. The amino acid sequences of GmASNase were submitted to the Swiss-Model online server in order to simulate its tertiary structure (Figure 2B) using the *Helicobacter pylori* L-asparaginase structure (PDB: 2wlt) as a template. Three-dimensional studies on GmASNase monomer revealed 30.13% of α-helix and 17.63% of β-strands (Figure 2C), which is similar to the structure of most L-asparaginases, such as HpA (PDB ID: 2WLT) from *Helicobacter pylori*, EcAII (PDB ID: 6PA3) from *E. coli*, TkA (PDB ID: 5OT0) from *T**hermococcus*
*kodakarensis*, and PhA (PDB ID: 1WLS) from *Pyrococcus horikoshii*. 

### 3.3. Expression and Purification of Mycobacterium Gordonae L-Asparaginase in E. coli

The purified GmASNase showed a single protein band between 25 and 35 kDa on SDS-PAGE, as shown in Figure 3, which is consistent with the predicted value of 31.308 kDa. The crude enzyme and the purification results are shown in Table 1. The specific activity of purified GmASNase was 486.65 IU mg^−1^, with a purification fold of 31.29 and a total yield of 42.71%.

### 3.4. Effects of pH and Temperature on the Activity and Stability of GmASNase

The activity of purified GmASNase was measured at various temperatures (Figure 4A). The optimum reaction temperature was 50 °C. Thermal stability was determined by measuring the residual activity of the enzyme after incubation at different temperatures (4–50 °C) for 50 min (Figure 4B). GmASNase displayed remarkable thermostability over the range of 4 to 35 °C, and the residual activity of GmASNase decreased significantly after 50 min of incubation at 40 and 45 °C. Particularly, the enzyme activity of L-asparaginase was lost at 50 °C for 10 min. The activity of GmASNase was measured under different pHs to determine its optimum reaction pH. As shown in Figure 4C, GmASNase had the enzymatic activity in the range of pH 6.0–12.0, with the highest catalytic activity at pH 9.0. In addition, GmASNase was incubated in buffers with different pH buffer for 24 h at 4℃, and reversed to the optimum pH for activity determination for its pH stability assay. The results showed that GmASNase was stable at pH 5.0–11.0 and still retained more than 80% of its residual activity after incubation at pH 6.0–11.0 for 24 h (Figure 4D).

### 3.5. Effect of Metal Ions and Inhibitors on GmASNase Activity

The effects of different metal ions and inhibitors on GmASNase were studied (Table 2). After incubation with Mn^2+^, Mg^2+^, and Ni^2+^ at a concentration of 1 mM, the activity of GmASNase clearly increased by 46.92%, 92.52%, and 59.71%, respectively. In particular, Mg^2+^ increased the relative activity by approximately twice that of the control, while the addition of Cu^2+^, Fe^2+^, and Fe^3+^ led to a reduction in GmASNase activity to 77.30%, 72.08%, and 79.51%, respectively. Other metal ions had little effect on the activity of the enzyme. Moreover, the effect of the inhibitors EDTA and SDS on GmASNase activity were also studied. EDTA had a slight effect on the activity of GmASNase. However, GmASNase activity was completely inhibited by SDS at a concentration of 1 mM.

### 3.6. Substrate Specificity Assay and Kinetic Parameters of GmASNase

GmASNase only had catalytic activity towards L-asparagine and D-asparagine, and showed no catalytic activity to other substrates (Figure 5A). Additionally, its activity towards D-asparagine was 22% of its L-asparagine activity. The values of the kinetic parameters Km and Kcat of GmASNase towards L-asparagine were calculated according to the Lineweaver–Burk double reciprocal plot (Figure 5B). The values of the kinetic parameters Km, kcat, and kcat/Km of GmASNase towards L-asparagine were determined to be 6.025 mM, 11,864.71 min^−1^, and 1969.25 mM^−1^min^−1^, respectively.

### 3.7. Application of GmASNase in the Mitigation of Acrylamide in Fried Potato Chips

A repetition trial using fried potato chips resulted in acrylamide levels of 0.559 ± 0.110 mg kg^−1^ for the control, and 0.195 ± 0.016 mg kg^−1^ for the enzyme-treated sample, with 40 IU mL^−1^ at 37 ℃ for 30 min, corresponding to a 65.09 ± 6.40% reduction in acrylamide (Appendix A). Hence, GmASNase could effectively inhibit the production of acrylamide in fried foods.

## 4. Discussion

L-asparaginases have attracted a significant amount of attention owing to their clinical use in acute lymphoblastic leukemia treatment and their range of applications in the food industry in order to reduce acrylamide formation in fried and baked foods, especially in potato chips, coffee, and cookies [36,37]. It should be noted that the well-known L-asparaginases from *A. oryzae* and *A. niger* have been commercially approved use in the food industry for acrylamide mitigation. Unfortunately, the pH and temperature characteristics of *A. oryzae* and *A. niger* L-asparaginase limit its range of industrial applications. Hence, it is necessary to explore new microbial species for the production of L-asparaginase with a wide pH and temperature stability range. So far, L-asparaginases from actinomycete species have rarely been reported.

In order to alleviate this issue, we identified a novel *Mycobacterium gordonae* L-asparaginase (GmASNase) that encoded the protein with a molecular mass of approximately 32 kDa consisting of 312 amino acid residues. GmASNase shared the highest similarity with some *mycobacterium* L-asparaginases by multiple sequence alignment, indicating that it is a novel *mycobacterium* L-asparaginase (Figure 1). Surprisingly, GmASNase was found in two branches associated with *Streptomyces* and *Nocardia* L-asparaginases, which were both derived from actinomycetes. However, there were obviously significant differences at the nucleotide level, indicating the divergence among the organisms. To further identify the structural differences with other L-asparaginases, a 3D structural model of GmASNase was predicted (Figure 2B). GmASNase was a tetramer composed of four identical monomers, which could also be said to be a dimer of two intimate dimers. There was a highly flexible “β-hairpin” structure in the *N*-terminal domain, which is responsible for substrate entry and product release in an open and closed manner [38]. The subunit of GmASNase consists of an *N*-terminal and a C-terminal domain connected by a linker formed by 173–200 residues (Figure 2C). Importantly, GmASNase had a maximum of 32% identity and a minimum of 23% identity with template L-asparaginases in the Protein Data Bank (PDB) database, indicating that it was a novel L-asparaginase.

To look into the activity of GmASNase, *E. coli* BL21 (DE3) was selected as its host cell. The activity of GmASNase reached 68.93 IU mL^−1^, which is more than double that of *Streptomyces* sp. L-asparaginase (26.67 IU mL^−1^) [39] and more than triple that of *S. phaeochromogenes* L-asparaginase (19.2 IU mL^−1^) [24]. As expected, the specific activity of purified GmASNase was as high as 486.65 IU mg^−1^, which is 31.30 times that of the crude enzyme (15.55 IU mg^−1^), and even higher than those of L-asparaginases from many other sources, such as *P. fluorescens* (26 IU mg^−1^) [40], *S. brollosae* (76.671 IU mg^−1^) [41], *A. terreus* (339.34 IU mg^−1^) [42], *B. licheniformis* (36.08 IU mg^−1^) [43], and *B. amyloliquefaciens* (4.75 IU mg^−1^) [31]. However, it is still lower than L-asparaginases from *B. sonorensis* (4438.62 IU mg^−1^) [44], *P. furiosus* (11,203.5 IU mg^−1^) [45], and *B. altitudinis* (800 IU mg^−1^) [46]. SDS-PAGE results showed a single protein band between 25 and 35 kDa for GmASNase, which is consistent with the predicted value. This result is similar to those for L-asparaginases from *P. fluorescens* (35 kDa) [40], *B. altitudinis* (35 kDa) [46], *T. kodakarensis* (32.6 kDa) [47], and *P. furiosus* (33.66 kDa) [45].

Generally, the optimum pHs of most L-asparaginases are between 7.0 and 8.0 (Appendix A). However, GmASNase displayed an alkaline maximum pH of 9.0, which is consistent with *S. gulbargensis* L-asparaginase [26] and *Streptomyces* sp. L-asparaginase [48]. Surprisingly, GmASNase showed an outstanding stability over a wide pH range of 5.0–11.0 and even showed more than 80% residual activity at pH 6.0–11.0 after 24 h of incubation. Studies show that L-asparaginases from actinomycete *A.*
*bacterium* [49], *S.*
*gulbargensis* [26], and *S.* spPKD2 [48] are only stable under alkaline conditions, and L-asparaginases from *A. oryzae* and *A. niger* L-asparaginases are only stable in the pH range of pH 6.0–9.0 and pH 5.0–9.0. In our study, GmASNase was stable in both acidic and alkaline conditions, which suggests that it would be suitable for high-pH baked foodstuffs such as corn flakes, crackers, and soda crackers.

Most L-asparaginases maintain their highest catalytic activity at 30–40 °C (Appendix A). However, GmASNase reached its maximum activity at 50 °C, similar to that of L-asparaginases from *P. furiosus* [45] and *S. noursei* [50], but obviously lower than that of L-asparaginase from thermophile (*T. kodakarensis*), whose highest catalytic activity was at 85 °C [47]. GmASNase was quite stable at 4 to 35 °C, approximately 50% of its residual enzyme activity could be retained after 30 min at 40 °C, and a significant loss of stability was recorded with the increasing of the temperature. However, L-asparaginase from *A. bacterium* [49] is only stable from 4 to 10 °C. In addition, L-asparaginase from *Acinetobacter soli* shows poor thermostability with a half-life at 40 °C of 9.63 min [10]. These results suggest that GmASNase is thermally stable.

Enzymes are protected against thermal denaturation by metal ions at elevated temperatures, which can maintain the active configuration of enzymes [41]. The Mg^2+^ co-factors can activate L-asparaginases by first activating the substrate and then combining it with the complex enzyme substrate to complete the enzyme-catalyzed hydrolysis reaction [21]. In our study, Mg^2+^, Mn^2+^, and Ni^2+^ played active roles in activating GmASNase, which were identical to those of the L-asparaginases from *A. bacterium* [49], *B. amyloliquefaciens* [31], *T. zilligii* [12], *Cobetia amphilecti* [51], and *P. fluorescens* [40]. However, divalent metals do not affect the enzymatic activity of L-asparaginase from *T. kodakarensis* [47]. EDTA inhibits the activity of L-asparaginase from *Fusarium culmorum* ASP-87 by 88% [52]. Meanwhile, the insensitivity of GmASNase in the presence of EDTA suggested that it might not be a metalloprotein. In addition, the anionic surfactant SDS completely inhibited the activity of GmASNase. Janakiraman also reported a similar finding that SDS inhibited the production of L-asparaginase in *F. culmorum* ASP-87 [52]. Mg^2+^, Mn^2+^, and Ni^2+^ could be introduced into the late fermentation process of GmASNase to optimize the fermentation conditions, while Fe^2+^, Fe^3+^, and Cu^2+^ should be avoided as much as possible.

GmASNase exhibited strict specificity towards L-asparagine (100%), but low activity on D-asparagine (22%) (Figure 5A). Ran et al. found that the high catalytic activity of D-asparaginase might be associated with the particularity of the substrate binding pocket by multipoint mutation and structure parsing [38]. Moreover, GmASNase displayed a Km value of 6.025 mM, which is lower than those of many other L-asparaginases from different species (Appendix A), such as *Rhizobium etli* (8.9 mM), *P. furiosus* (12 mM), *T. gammatolerans* EJ3 (10 mM), and *P. oyzihabitans* (10.0 mM). However, it is still higher than L-asparaginases from *E. chrysanthemi* and *E. coli*. This strict substrate specificity offers GmASNase as a potential candidate for applications. 

As a threat to the health of human beings, acrylamide has been produced in fried food products. L-asparaginases have been used in fried potato chips to verify their abilities in degrading acrylamide. The reduction rates of most L-asparaginases on acrylamide in fried potato chips vary from 55 to 96% (Appendix A). The content of acrylamide decreased by 65.09% when adding GmASNase in advance, which is higher than that of *Acinetobacter soli* (55.9%) [10]. This positive effect may be due to the fact that GmASNase hydrolyzed L-asparagine (the crucial precursor of acrylamide) in potato chips, indicating that GmASNase is an efficient agent in the food processing industry.

In conclusion, GmASNase was successfully heterogeneously expressed and functionally characterized. The results suggested that GmASNase displayed stability at a wide pH range, activity at various temperatures, and high specificity to L-asparagine. Additionally, GmASNase was also proven to have the ability to reduce the acrylamide content in fried potato chips. Our research explored GmASNase a future substitute of commercial L-asparaginases for food processing applications. 

## Figures and Tables

**Figure 1 foods-10-02819-f001:**
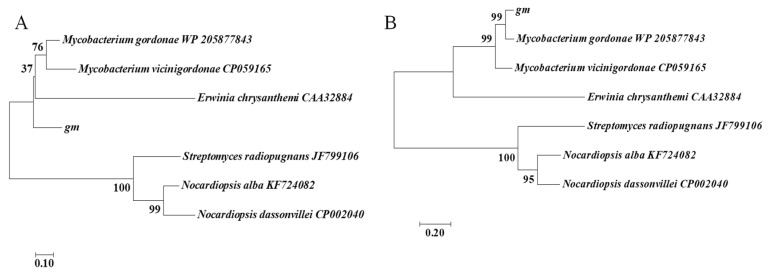
(**A**) Phylogenetic tree analysis of nucleotide sequences of *Mycobacterium gordonae* L-asparaginase. (**B**) Phylogenetic tree analysis of amino acid sequences in *Mycobacterium gordonae* L-asparaginase.

**Figure 2 foods-10-02819-f002:**
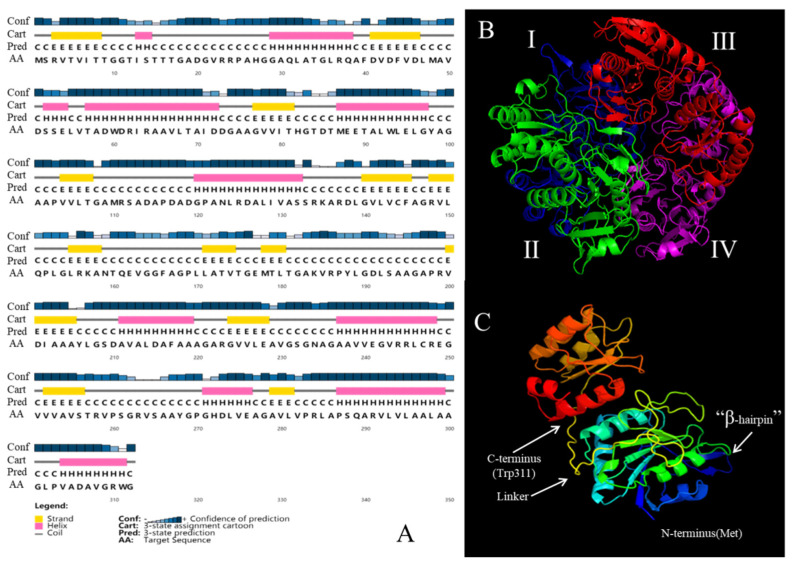
(**A**) Secondary structure prediction for the deduced amino acid sequence of L-asparaginase gene from *Mycobacterium gordonae* by PSIPRED. (**B**) Three-dimensional model prediction of L-asparaginase protein from *Mycobacterium gordonae*; I (blues); II (greens); III (reds); IV (magentas). (**C**) Dimer structure model of L-asparaginase protein from *Mycobacterium gordonae*.

**Figure 3 foods-10-02819-f003:**
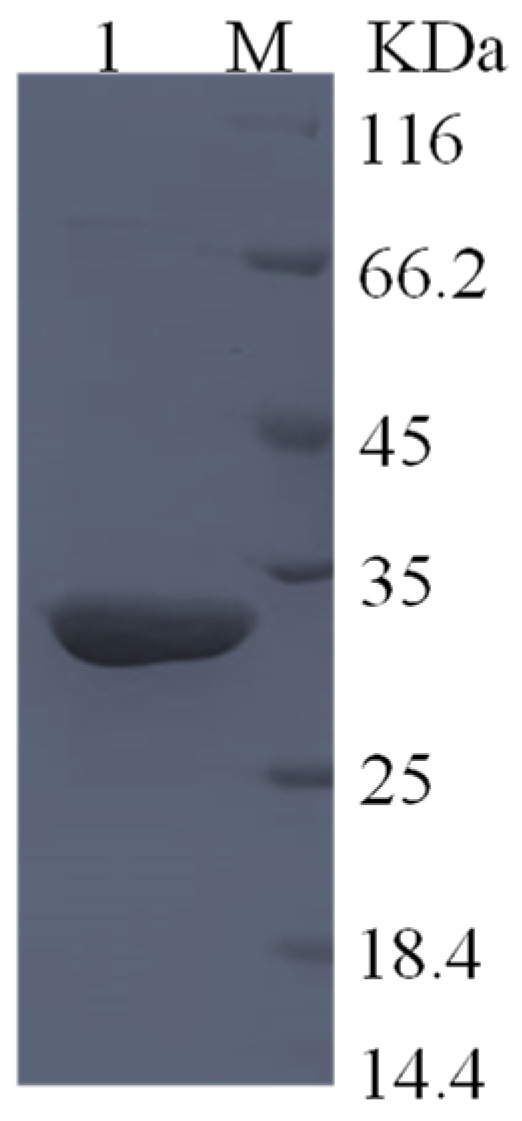
SDS-PAGE analysis of GmASNase; M, protein marker; lane 1, purified GmASNase.

**Figure 4 foods-10-02819-f004:**
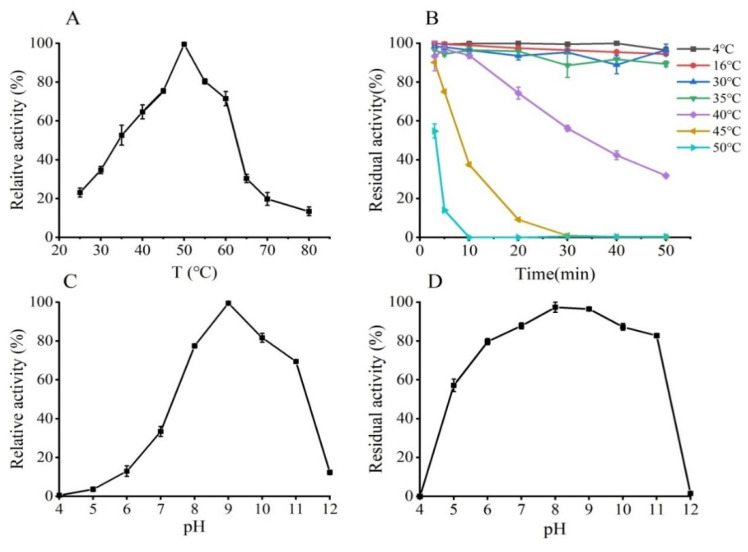
Effects of temperatures (**A**,**B**) and pHs (**C**,**D**) on the activity and stability of GmASNase. Data are expressed as mean values ± SD of three independent experiments.

**Figure 5 foods-10-02819-f005:**
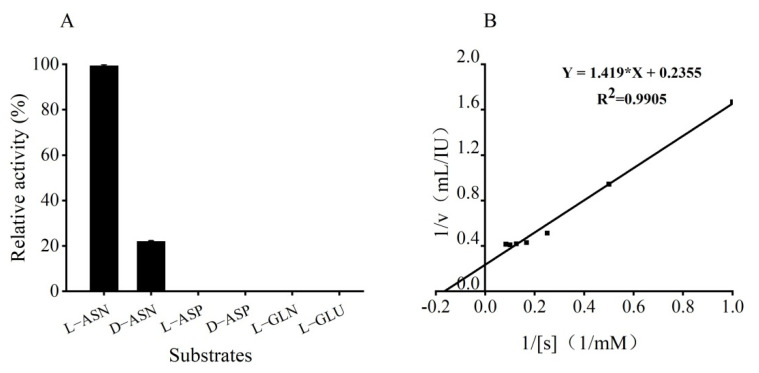
Substrate specificity assay (**A**) and kinetic parameters (**B**) of GmASNase. Data are expressed as mean values ± SD of three independent experiments; L-ASN (L-asparagine); D-ASN (D-asparagine); L-ASP (L-aspartic acid); D-ASP (D-aspartic acid); L-GLN (L-glutamine); L-GLU (L-glutamic acid).

**Table 1 foods-10-02819-t001:** Separation and purification of GmASNase.

Purification Steps	Total Activity (IU)	Total Protein (mg)	Specific Activity (IU mg^−1^)	Purification Fold	Yield (%)
Crude enzyme	3110.86	200.04	15.55	1	100
Nickel affinity purified GmASNase	1328.57	2.73	486.65	31.29	42.71

**Table 2 foods-10-02819-t002:** Effects of metal ions and inhibitors on GmASNase activity.

Metal Ions and Inhibitors	Relative Activity (%)
Control	100
Mn^2+^	147 ± 10
Zn^2+^	103 ± 4
K^+^	94 ± 3
Na^+^	93 ± 1
Al^3+^	84 ± 2
Ca^2+^	89 ± 6
Mg^2+^	193 ± 29
Cu^2+^	77 ± 4
Ni^2+^	160 ± 8
Fe^2+^	72 ± 2
Fe^3+^	80 ± 3
SDS	0
EDTA	84 ± 1

Data are expressed as mean values ± SD of three independent experiments; SDS (sodium dodecyl sulfate); EDTA (ethylene diamine tetraacetic acid).

## Data Availability

Data is contained within this article and Appendix A.

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
