# Peer review of "Characterization of a Novel L-Asparaginase from Mycobacterium gordonae with Acrylamide Mitigation Potential"

_foods, 2021, doi:10.3390/foods10112819_

Round 1

Reviewer 1 Report

All my previous comments have been addressed.

Author Response

Manuscript ID: foods-1451040
Title: "Characterization of a novel L-asparaginase from Mycobacterium gordonae with acrylamide mitigation potential"
Author(s): Huibing Chi, Meirong Chen, Linshu Jiao, Zhaoxin Lu, Xiaomei Bie, Haizhen Zhao, and Fengxia Lu*

Dear Reviewers:

Thank you very much for the careful reading of our manuscript entitled "Characterization of a novel L-asparaginase from Mycobacterium gordonae with acrylamide mitigation potential", and for the opportunity to submit a revised version. We sincerely appreciate for providing valuable suggestions and comments that have greatly helped us to improve the manuscript.

We apologize for the writing which is not rigorous enough. The revised manuscript has checked by "MDPI English editing" to ensure that there are no issues related to language or grammar.

Attached, please find our point-by-point response to the reviewers’ comments. We would be most grateful if you could consider the revised version for publication foods.

We look forward to your response.

Sincerely,

Prof. Fengxia Lu

College of Food Science & Technology, Nanjing Agricultural University

Nanjing 210095, China

Response to Reviewer 1 Comments

Comments

All my previous comments have been addressed.

Response: We appreciate the referee’s comments again.

Response to Reviewer 2 Comments

Comments

The authors made the changes suggested by the reviewer. However, most of the sentences added need an enghlish revision. In this sense, I recommend corrections (english revision).

Response: We apologize for the writing which is not rigorous enough. The revised manuscript has checked by "MDPI English editing" to ensure that there are no issues related to language or grammar.

Reviewer 2 Report

The authors made the changes suggested by the reviewer. However, most of the sentences added need an enghlish revision. In this sense, I recommend corrections (english revision).

Author Response

Manuscript ID: foods-1451040
Title: "Characterization of a novel L-asparaginase from Mycobacterium gordonae with acrylamide mitigation potential"
Author(s): Huibing Chi, Meirong Chen, Linshu Jiao, Zhaoxin Lu, Xiaomei Bie, Haizhen Zhao, and Fengxia Lu*

Dear Reviewers:

Thank you very much for the careful reading of our manuscript entitled "Characterization of a novel L-asparaginase from Mycobacterium gordonae with acrylamide mitigation potential", and for the opportunity to submit a revised version. We sincerely appreciate for providing valuable suggestions and comments that have greatly helped us to improve the manuscript.

We apologize for the writing which is not rigorous enough. The revised manuscript has checked by "MDPI English editing" to ensure that there are no issues related to language or grammar.

Attached, please find our point-by-point response to the reviewers’ comments. We would be most grateful if you could consider the revised version for publication foods.

We look forward to your response.

Sincerely,

Prof. Fengxia Lu

College of Food Science & Technology, Nanjing Agricultural University

Nanjing 210095, China

Response to Reviewer 1 Comments

Comments

All my previous comments have been addressed.

Response: We appreciate the referee’s comments again.

Response to Reviewer 2 Comments

Comments

The authors made the changes suggested by the reviewer. However, most of the sentences added need an enghlish revision. In this sense, I recommend corrections (english revision).

Response: We apologize for the writing which is not rigorous enough. The revised manuscript has checked by "MDPI English editing" to ensure that there are no issues related to language or grammar.

This manuscript is a resubmission of an earlier submission. The following is a list of the peer review reports and author responses from that submission.

Round 1

Reviewer 1 Report

The paper by Chi et al. is a nice example of the characterization of a new agent preventing the formation of acrylamide in food industry. In my opinion the paper is correctly written, the experiments are well conducted and the results seem to support the research. Some comments:

- industrial application: what would be the economic cost of applying the agent to food industry?

- real application: the authors prepare potato chips using different steps (page 4). Would these steps be applicable to real food industry? How would all the steps be in a real commercial food (e.g. frozen French fries) available to the public? How would all the steps be in a real food (e.g. raw potatoes) bought in any local market? In my opinion if the authors want to show that GmASNase is a potential candidate for applications in the food industry, it should be applied in real conditions, not only in lab conditions

- Why 40 IU mL-1 of enzyme solution are used? Is this the optimum value?

- figure 4, error bars in the different effects. How many replicates did the authors made? In which conditions (same day, different day, same analyst …)?

- table 2, significant figures. Probably it does not make sense to include two decimal places. In, e.g., 146.92 +- 10.37, if there is an uncertainty of the order of 10, I do not see the point to remark the .92. Probably here 147+-10 would be enough.

- section 3.7, ‘A repetition trial in fried potato chips…’. If the authors conducted a repetition trial, I would be expecting a +- in the value of 65.09% corresponding to the reduction in acrylamide.

Reviewer 2 Report

In this work, the authors characterized a novel L-asparaginase from Mycobacterium gordonae (GmASNase) cloned and expressed in Escherichia coli BL21 (DE3). The enzyme activity in different temperatures and pH values was investigated, as the ability to mitigate the acrylamide formation in potato chips. The manuscript is interesting, but some concepts need to be improved. For instance, the authors characterized a novel glutaminase-free L-asparaginase and proposed their application in the food industry due to its acrylamide mitigation potential. However, the potential of this enzyme to be used in clinical applications is several times highlighted, due to its zero glutaminase activity. Why the authors choose the food industry as a potential application for this enzyme? Why not a clinical application? I think this is not clear. Otherwise, the article doesn’t fit in the journal subject.  In this sense, I recommend major corrections . Moreover, I suggest other modifications to improve the paper quality.

Abstract:

  1. Line 17: “GmASNase was 17 found to be a tetramer with monomeric size of 32 kDa, sharing only 32% structural identity with 18 other reported L-asparaginases in Protein Data Bank database”. Which L-asparaginases? Add this information to the manuscript.

Introduction:

  1. Line 40: “L-asparaginase has been found in animals, plants and microorganisms. However, only microbial L-asparaginases are of commercial significance” Why only microbial L-asparaginases are of commercial significance? Add this information to the manuscript.
  2. Line 41: “To date, the properties of L-asparaginases from a variety of microorganisms have been characterized [13, 14], especially the best-known bacterial-type L-asparaginases from Erwinia chrysanthemi and Escherichia coli are currently in clinical use” Revise this sentence.
  3. Line 46: “Therefore, it is desirable to achieve glutaminase-free L-asparaginase for quality and safety.” This work is focus on food industry. In this field is necessary a glutaminase-free L-asparaginase?
  4. Line 48: L-asparaginases produced by Aspergillus oryzae and Aspergillus niger have only a catalytic activity for asparagine or they also catalyze glutamine? Why the L-asparaginase produced by Aspergillus oryzae and Aspergillus niger are not used in clinic applications?
  5. Line 68-71: The main results of the work developed are already disclosed in the introduction. Revise this part of the introduction to avoid this information.

Materials and methods:

  1. Line 92-94: “Protein expression was induced by adding IPTG at a final concentration of 100 mg mL-1, followed by incubation at 16°C for 20 h.” The induction of protein expression and fermentation occurred in these conditions based in previous reported works? References must be added.
  2. Line 98: “The recombinant GmASNase was checked by 12% SDS-PAGE.” More details on the SDS-PAGE protocol must be added to the manuscript. For instance, the protein concentration loaded, how the gel was stained, among others.
  3. Line 117-120: The catatylic of GmASNase was detected in different pH values. Totally, five different buffers were used and composed with different salts. Why the authors selected these buffers? Why didn’t the authors use buffers with a higher range of pHs in order to reduce the number of buffers used avoiding the variability of the salts used? This information must be added to the manuscript.
  4. Line 122-124: To determine the GmASNase relative activity at each pH value, after 24h the enzyme was reversed to the optimum pH for activity determination. Why? This procedure as to be clarified.
  5. Line 146: All the periods of time are in min, excepted “0.5h”. Change this value to min to maintain the formation.

Results:

  1. Line 200-202: “The amino acid sequences of GmAsNase were submitted to the Swiss-Model online server to simulate its tertiary structure (Figure 2B) using Helicobacter pylori L-asparaginase structure (PDB: 2wlt) as template.” Why Helicobacter pylori L-asparaginase structure was selected as the template?
  2. Line 204: “most of L-asparaginases”. Which asparaginases? Add this information.
  3. Line 233: Figure 3 has “M, protein marker” Although is clear what is the protein marker, in the image is not present any “M”. Change the Figure according to the image or the contrary.

Discussion:

  1. Line 287: “… be cytotoxicity for its L-glutaminase activity in clinic.”; Line 367: “Therefore, the property of zero glutaminase activity may make GmASNase a new drug for the treatment of cancer or leukemia.” The glutaminase activity is toxic for food applications? Clinical applications are highlighted several times. To publish this paper in Food, this idea must be clarified. If the glutaminase activity is not toxic for food applications why this information must be reduced. On the other hand, if the authors decided to maintain this main idea, maybe the Food Journal is not the proper choice to publish this paper.